# AdaMM: Adaptive Object Movement and Motion Tracking in Hierarchical Edge Computing System

**DOI:** 10.3390/s21124089

**Published:** 2021-06-14

**Authors:** Jingyeom Kim, Joohyung Lee, Taeyeon Kim

**Affiliations:** 1School of Computing, Gachon University, Seongnam 13120, Korea; kimo1113@gachon.ac.kr; 2Electronics and Telecommunications Research Institute (ETRI), Daejeon 34129, Korea; tykim@etri.re.kr

**Keywords:** EdgeAI, hierarchical edge computing, deep learning, object detection and tracking, software implementation

## Abstract

This paper presents a novel adaptive object movement and motion tracking (AdaMM) framework in a hierarchical edge computing system for achieving GPU memory footprint reduction of deep learning (DL)-based video surveillance services. DL-based object movement and motion tracking requires a significant amount of resources, such as (1) GPU processing power for the inference phase and (2) GPU memory for model loading. Despite the absence of an object in the video, if the DL model is loaded, the GPU memory must be kept allocated for the loaded model. Moreover, in several cases, video surveillance tries to capture events that rarely occur (e.g., abnormal object behaviors); therefore, such standby GPU memory might be easily wasted. To alleviate this problem, the proposed AdaMM framework categorizes the tasks used for the object movement and motion tracking procedure in an increasing order of the required processing and memory resources as task (1) frame difference calculation, task (2) object detection, and task (3) object motion and movement tracking. The proposed framework aims to adaptively release the unnecessary standby object motion and movement tracking model to save GPU memory by utilizing light tasks, such as frame difference calculation and object detection in a hierarchical manner. Consequently, object movement and motion tracking are adaptively triggered if the object is detected within the specified threshold time; otherwise, the GPU memory for the model of task (3) can be released. Moreover, object detection is also adaptively performed if the frame difference over time is greater than the specified threshold. We implemented the proposed AdaMM framework using commercial edge devices by considering a three-tier system, such as the 1st edge node for both tasks (1) and (2), the 2nd edge node for task (3), and the cloud for sending a push alarm. A measurement-based experiment reveals that the proposed framework achieves a maximum GPU memory reduction of 76.8% compared to the baseline system, while requiring a 2680 ms delay for loading the model for object movement and motion tracking.

## 1. Introduction

The recent proliferation of deep learning (DL) technology has led to an unprecedented boom of artificial intelligence (AI) in our lives [1,2]. Moreover, with the widespread use of approximately 3 billion smartphones, 7 billion Internet of things (IoT) devices, and single board computers (e.g., Raspberry Pi), AI-based applications have been rapidly popularized and researched in both video surveillance and crowd sourcing systems, which are broadly classified as video surveillance services [3,4]. In AI-assisted video surveillance services, to detect certain events or target objects with the aim of context awareness (e.g., patient monitoring in healthcare service, driving accident detection for connected cars), a large amount of image data captured at the end devices (e.g., smartphones, IoT devices) is streamed to the centralized node (i.e., cloud server) for DL model inference owing to the limited computing capability of end devices. Moreover, such a cloud-centric approach is no longer sustainable considering the following challenges. First, streaming high-quality image data to the cloud for inference burdens the backbone networks in the centralized cloud. This eventually incurs unacceptable latency in addition to a long propagation delay to the cloud. Second, transmitting raw image data to the cloud raises privacy concerns in cases where some data owners are unwilling to expose privacy-sensitive image data to the cloud.

To tackle the aforementioned challenges, edge computing-based AI architecture, known as edge AI or edge intelligence, has been researched and proposed. In such an architecture, the edge computing node is owned by the data owners and is physically close to the end devices. In the survey work of Reference [2,5], the architecture of edge AI for DL model inference is four-fold: (1) edge server-based, (2) edge device-based, (3) edge server-edge device, and (4) edge server-cloud. In the (1) edge server-based architecture, the inference is performed on a nearby edge server, and its result is sent back to the edge device from the edge server. Herein, the performance is determined by the network status between the edge server and edge device. In the (2) edge device-based architecture, the inference is performed locally at the edge device with the DL model obtained from a nearby edge server. During the inference phase, the edge device does not communicate with the edge server. Therefore, the inference process is reliable, but it requires a considerable amount of computing resources (e.g., GPU and CPU) on the edge device. Unlike the previous two architectures, in the (3) edge server-edge device architecture, the DL model is stored on both the edge server and edge device. The edge device partitions the DL model into several parts based on the network bandwidth and end-device resources. Then, the edge device runs a part of the DL model up to a specific layer and sends the intermediate result to the edge server. Subsequently, the edge server runs the remaining DL model and sends the final result to the edge device. This architecture is more stable and flexible than the (1) edge server-based architecture, but it is complex for practical implementation. Lastly, the (4) edge server-cloud architecture is similar to the (3) edge server-edge device architecture on the count that inference is performed separately. However, the inference is performed on the edge server and cloud data center. The edge device collects data, and the edge server performs intermediate inference and sends the result to the cloud data center. Finally, the cloud data center executes the inference and sends the final result to the edge device. Here, the performance is dependent on the network connection status because it involves considerable communication.

More recently, the aforementioned edge computing-based AI architectures have been actively investigated and commercialized for video surveillance services in the industry [6,7,8]. An example of commercialization is NTT Docomo’s “Edge AI Platform” service and KDDI’s “Video X 5G” service [9,10]. In Reference [9], NTT’s Edge AI platform service is designed with an Edge AI Box as the edge device, which is equipped with a 4G/5G modem, CPU, and GPU. Such Edge AI Box is closely located at the enterprise, and it processes the DL inference locally so that it is not necessary to directly send a large video stream to the cloud for AI-assisted video surveillance. Consequently, the Edge AI Box in an organization can not only reduce latency and traffic but also alleviate privacy threats. In Reference [10], KDDI’s Video X 5G service, which is similar to that of NTT’s, but considers different architecture options, adopts a mobile edge computing (MEC) architecture. The main idea is to run AI-assisted video surveillance services, such as AI image analysis, face recognition, and high-definition streaming on the MEC server, which is owned by the service provider and not located within the organization. This leads to space and service cost savings for the organization.

Several approaches were proposed in Reference [11,12,13,14,15,16] for performing inference and improving its speed on the resource-constrained edge server (or edge device). Specifically, research in Reference [11,12] proposed the concept of edge caching, which is caching and reusing the previous inference result. In Reference [13,14], researchers studied the impact of early existing, i.e., using the result of the early layer and not the final one to obtain an inference layer. Furthermore, in Reference [15,16], model compression was adopted to reduce the complexity and computation resources of the DL model, and the impact of hierarchical edge computing architecture was studied. Recently, Li et al. [17] developed a joint accuracy and latency-aware deep neural network (DNN) framework that partitions DNNs and operates on edge devices and cloud servers with a partitioned DNN model. It improves the DNN inference speed while guaranteeing model accuracy. The dynamic adaptive DNN surgery (DADS) scheme proposed by Hu et al. [18] aims to optimally partition a DNN network based on periodic monitored network conditions. Alam et al., in Reference [19], proposed an efficient edge computing architecture for an unmanned aerial vehicle (UAV) scenario to reduce latency and network traffic while detecting objects and abnormal events. The proposed architecture aims to filter the frames that are not of interest in the video at the edge device while sending only the frames that need to be processed for the inference, to the cloud server. In addition, Lee et al. [20] proposed a novel and robust object detection and tracking method, which consists of three modules: object detection, tracking and learning. It can track the interest object accurately compared to traditional method while focusing on reducing GPU processing power and improving motion tracking accuracy.

Nevertheless, to the best of our knowledge, most research work is limited to accelerating the inference speeds by reducing the GPU processing power. DL-based inference for object movement and motion tracking in video surveillance services requires considerable GPU memory for model loading and GPU processing power for the DL inference. Despite the absence of the object in the video, if the DL model is loaded, the GPU memory must be kept allocated for the loaded model. Moreover, several cases of video surveillance try to capture events that rarely occur (e.g., abnormal object behaviors); therefore, such standby GPU memory might be easily wasted. However, only a few studies on the edge computing system tackle the GPU memory issue [21,22], and they are confined to model compression for saving GPU memory. Moreover, these studies do not consider the hierarchical edge computing system and the practical implementation for video surveillance services.

In this paper, we propose a novel adaptive object movement and motion tracking (AdaMM) framework in a hierarchical edge computing system for achieving GPU memory footprint reduction of DL-based video surveillance services. To alleviate the aforementioned challenges, the proposed framework aims to adaptively release the unnecessary standby object motion and movement tracking model to save GPU memory by checking whether there are objects of interest in the video within a specified threshold interval. The proposed framework can be applied to various services, such as video surveillance systems that monitor unusual behavior in public places to ensure safety by using cameras on vehicles or drones. To our knowledge, the proposed AdaMM framework is the first study to (1) exploit dynamic the DNN model-release for the object movement and motion tracking, which requires heavy GPU memory usage for standby and (2) consider hierarchical edge computing architecture for adaptive object movement and motion tracking in the video surveillance services and its implementation on the commercial edge devices for measurement-based experiment. Specifically, most of the research works consider the always-on DNN model for the inference phase even though various video surveillance scenario try to capture events that rarely occur (e.g., abnormal object behaviors); therefore, such standby GPU memory for always-on object movement and motion tracking might be easily wasted, which we tackle with. Moreover, such distributed system architectures have been intensively researched in the area of distributed learning (e.g., federated learning) in order to alleviate privacy concerns and bottleneck of centralized training server [23,24]. Nevertheless, study on the distributed system architectures for the inference phase, which requires also both heavy GPU memory and GPU processing power, have not been sufficiently conducted compared to training phase. The reason behind this is that training phase requires more GPU computing power and memory usage than inference phase.

The detailed contributions of this study are summarized as follows:The proposed AdaMM framework considers the hierarchical edge computing system. The hierarchy of the proposed framework is three-fold: (1) 1st edge nodes, (2) 2nd edge nodes, and (3) cloud servers in an increasing order of computing resources (i.e., GPU and CPU). Moreover, the proposed AdaMM framework categorizes tasks as follows: (1) task (1)—object movement and motion tracking procedure, (2) task (2)—object detection, and task (3)—object motion and movement tracking, in an increasing order of the required processing and memory resources.The proposed AdaMM framework is designed to adaptively release the unnecessary standby object motion and movement tracking model so that GPU memory can be saved by utilizing light tasks, such as task (1) frame difference calculation and task (2) object detection, in a hierarchical manner. Accordingly, object movement and motion tracking are adaptively triggered if the object is detected within the specified threshold time; otherwise, the GPU memory for the model of task (3) can be released. Moreover, object detection is also performed adaptively if the frame difference over time is greater than the specified threshold.To do this efficiently in a hierarchical manner, the 2nd edge node executes task (3) object movement and motion tracking, and the 1st edge nodes that have extremely limited computing resources run light tasks, such as task (1) frame difference calculation and task (2) detecting objects to determine whether to trigger object movement and motion tracking at the 2nd edge node. More specifically, if there is a large difference in the frames or any of the objects are recognized, then a message is sent to the 2nd edge node. Then, the 2nd edge node receives this message with the image frames and executes task (3) the object movement and motion tracking process. Finally, the 2nd edge node sends a message to the cloud server to trigger the notification process for the clients that are registered for the service (e.g., alert the registered clients regarding the dangerous situation of the target objects (e.g., people)). In the last tier, the cloud server allows users to view the video captured by the 1st edge nodes and the notification generated by the 2nd edge nodes in real time.As for the implementation details, we designed a message format for communication with the 1st edge nodes, 2nd edge nodes, and cloud servers. Further, we implemented the proposed AdaMM framework using commercial edge devices, such as NVIDIA Jeston Nano and the Android platform. A measurement-based experiment reveals that the proposed framework achieves a maximum GPU memory reduction of approximately 76.8% compared to the baseline system, while requiring a 2680 ms delay for loading the model for object movement and motion tracking.

The remainder of this paper is organized as follows. In Section 2, the proposed framework is described. Implementation of the framework is discussed in Section 3. Section 4 presents the results. Finally, Section 5 concludes the paper.

## 2. Adaptive Object Movement and Motion Tracking (AdaMM) Framework

In this section, we first describe the proposed AdaMM framework, which is illustrated in Figure 1. It is comprised of three layers of network nodes:**1st Layer**: 1st edge node (i.e., edge device) is responsible for tasks (1) calculating the frame difference and (2) running the object detection module.**2nd Layer**: 2nd edge node (i.e., edge server) is responsible for tasks (3) running control module and (4) motion tracking module and sending trigger messages to 3rd Layer.**3rd Layer**: Cloud server is responsible for tasks (6) dispatching notifications to registered clients.

For simplicity, we consider that a single 1st edge node is attached to the 2nd edge node. It should be noted that, in the proposed framework, multiple 1st edge nodes can be easily deployed. Specifically, the 1st edge node, which is directly connected to the IP camera, obtains the input video frames from the connected camera and is responsible for tasks (1) calculating the frame difference and (2) running the object detection module. At the 1st edge node, firstly, task (1) calculates the difference between the video frames, and when the difference exceeds the threshold, task (2) object detection module is triggered. The object detection module determines whether the 1st edge node should trigger object movement and motion tracking on the 2nd edge node. If the decision to trigger is made, the 1st edge node sends a trigger message along with image frames to the 2nd Layer (2nd edge node). The frames with the results of object detection (e.g., person bounding boxes) are sent to the 2nd edge node for object movement and motion tracking, where the 2nd edge node is in the same network as that of the 1st edge node. When the 2nd edge node receives the video frames from the 1st edge node, it runs task (3) control module that sets the threshold depending on the waiting time to receive the video frames from the 1st edge node. (3) The control module determines whether to trigger task (4) motion tracking module or stop it using a threshold. If the decision to trigger is made, the control module sends a trigger message with image frames to the motion tracking module, and, if the decision to stop is made, only the trigger message is sent. When the image frame is received from the control module, (4) the motion tracking module performs object movement and motion tracking. The motion tracking module’s task (5) sends trigger messages to the cloud server depending on the situation (e.g., a dangerous situation or abnormal behavior of a person). It should be noted that the proposed AdaMM framework can be generalized to all real-time video surveillance services; therefore, the proposed framework is not limited to a specific service scenario. The cloud server receives the trigger message from the 2nd edge node, and task (6) dispatches notifications to the registered clients (e.g., smartphones). In addition to the notification functionality, the proposed framework supports web-based monitoring services, which can provide real-time video streaming to registered users. Moreover, it can be supported by any device with a web-browser. Details of these modules of the proposed framework are described in subsequent sub-sections. For the reader’s convenience, we provide in Table 1 a list of math notations that we shall define and use in this paper.

### 2.1. Background

Compute unified device architecture (CUDA) is NVIDIA’s parallel computing platform and application programming interface that facilitates the use of a GPU for general-purpose processing [25]. Programs using CUDA have the following flow, as illustrated in Figure 2.

Copy data from main memory to the memory for GPU (i.e., GPU memory).CPU instructs the process to GPU.CPU instructs the process to GPU.Results from the GPU memory are copied to the main memory.

The parameter, current GPU usage utilization, means the percentage on usage actual of core, order 3. The current GPU memory utilization means the percentage on size of data that is copied from main memory to use GPU account for total size of GPU memory, it is regardless of GPU usage utilization. In TensorFlow, which is an end-to-end open source platform for machine learning, data is copied from main memory to GPU memory when the first GPU is used. Then, the GPU memory is held until the program terminates, even if the GPU is not executed. This causes a memory leak problem while copying data to GPU memory for other new tasks. To alleviate this problem, we propose the AdaMM framework that releases GPU memory whenever the GPU is not used.

### 2.2. Object Detection Module

The object detection module includes two steps: In the first step, the module calculates frame differences to reduce computation resources by deciding whether to run or skip object detection. To this end, the captured video frame image at arbitrary time *t* is converted to gray scale through a gray scaling function as in Reference [26], which is denoted as $gray_{t}\left(x,y\right)$ and is given by
(1)$$gray_{t}\left(x,y\right)=0.299\cdot f\left(x,y,R\right)+0.587\cdot f\left(x,y,G\right)+0.114\cdot f\left(x,y,B\right),$$
where f(x,y,R), f(x,y,G), and f(x,y,B) are the red channel, green channel, and blue channel pixel values in specific coordinates (x,y), respectively. Then, the summation of the difference value for every pixel between the frame at *t* and t−1 is calculated, and, if it exceeds threshold ϕ, then d(t) is 1; otherwise, 0.
(2)$$d\left(t\right)=\left\{\begin{array}{cc} 1 & \mathrm{if}\;|gray_{t}\left(x,y\right)-gray_{t-1}\left(x,y\right)|\;\geq \;\phi \\ 0 & \mathrm{otherwise} \end{array}\right.,$$
(3)$$D\left(t\right)=\sum_{xy}d\left(t\right).$$

By using Equation (Equation 3), if the frame difference, D(t), exceeds threshold $\theta_{f}$, then the 1st edge node runs the YOLO network to detect objects [27]. However, if D(t) does not exceed $\theta_{f}$, the frame is skipped to reduce the usage of computation resources. Here, the service provider can determine the values of ϕ and $\theta_{f}$ depending on their interests (e.g., minimizing computation resources or more frequent object detection for better accuracy). For instance, if the service provider intends to reduce the computation resources for the service, they can set ϕ to a higher value. In contrast, if they intend to conduct more frequent object detection in the image frames for improved accuracy, they can set ϕ to a lower value. In our experiment, threshold ϕ was set to 35 and $\theta_{f}$ to 50% of the frame size [11]. However, identifying the optimal settings of ϕ and $\theta_{f}$ is out of the scope of this paper.

The second step is to detect objects (e.g., person) using the YOLO network, which is a real-time object detection algorithm [27]. Particularly, in this study, the object detection module adopts YOLOv3-tiny, which is a very small model for constrained environments (In our experiment, we adopt YOLOv3-tiny, which was the latest version of YOLO during the development of the proposed system. It should be noted that the latest version of YOLO (e.g., YOLOv4) outperforms YOLOv3 in terms of recognition rates and processing speed but still has drawbacks in complex scenarios (i.e., poor performance during the night as a result of using mosaic data augmentation technique) [28]. Thus, YOLOv3 seems to be considered as a stabilized version, and it has been adopted in various academics and industries. Nevertheless, the proposed AdaMM framework is not be limited to specified version of YOLO, and it is easily replaced with other YOLO versions for performance improvement). Then, if the object detection module recognizes the presence of target objects in the frame, the video frame is delivered to the 2nd edge node using socket transmission.

### 2.3. Control Module

The control module controls whether to run or stop the motion tracking procedure, which depends on whether there were incoming video frames from the 1st edge node within the motion tracking threshold, $\theta_{m}$. The 2nd edge node runs the motion tracking and control modules, which is presented in Algorithm 1. Basically, the module receives video frames from the 1st edge node. The video frames from the 1st to the 2nd edge node are delivered in an event-driven manner. For instance, if there are any detected objects in the 1st edge node, the module sends the image frame to the 2nd edge node. Consequently, the motion tracking module can wait several seconds while holding GPU resources unless there are any detected objects in the 1st edge node. To reduce the unused GPU resource leakage, the control module sends a trigger message to stop the motion tracking procedure to release the GPU resources if the motion tracking threshold, $\theta_{m}$, has expired without incoming image frames. Otherwise, the control module sends the frame received from the 1st edge node to the motion tracking module to run the motion tracking procedure.
**Algorithm 1** Control Module 2nd Edge Node.**Input:** Motion tracking threshold $\theta_{m}$, queue size q
**while do**
    Put received frame to queue
    **if** Queue is empty **then**
        Record time of empty $t_{empty}$
        **if** $t_{empty}$ ≥ $\theta_{m}$ **then**
           Send trigger message with stop instruction to motion tracking module
        **end if**
    **else**
        Send received frame to motion tracking procedure with queue
    **end if**
**end while**


### 2.4. Motion Tracking Module

The motion tracking module runs or stops the motion tracking procedure depending on a trigger message from the control module. The motion tracking module stops the motion tracking procedure if the message is a control message to stop. However, if the message is an image frame, the motion tracking module runs the motion tracking procedure to track objects in the received image frame from the 1st edge node. Additionally, the motion tracking module sends a trigger message to the cloud server when a specific situation occurs. Such a message can contain a description of the situation.

### 2.5. Overall Procedure of AdaMM Framework for Multiple 1st Edge Nodes

Figure 3 illustrates a flow diagram of the proposed AdaMM framework, which can be generalized for multiple 1st edge nodes. The message format is illustrated in Figure 4. The overall framework includes the aforementioned object detection, control, and motion tracking modules. First, the 1st edge node obtains frames from the camera and runs the object detection module. Then, it calculates the frame difference between the current and the previous frame using Equation (Equation 3). If the frame difference exceeds threshold ϕ, the 1st edge node runs the object detection module. Unlike the case when the framework has a single edge node, in the case of multiple edge nodes, the 1st edge node sends the video frames with the 1st edge node ID for identification to the 2nd edge node.

Then, the 2nd edge node on receiving the frames with the messages, separately runs different modules and manages the queue based on the 1st edge node ID. Except for creating and managing queues based on the 1st edge node ID, it works as described in Section 2.3. For multiple 1st edge nodes, the cloud server must identify authorized users according to the 1st edge node IDs. Consequently, the cloud server sends notifications only to the authenticated users on the 1st edge node. The format of the message on each node is illustrated in Figure 4.

## 3. Implementation

In our test environment, the 1st edge node is implemented using Jetson Nano [29] (Based on measurement based experiment, we checked that the result of running motion tracking on Jetson Nano, represents about between 1.2 and 3 fps, which is not suitable for real time surveillance service. So, we adopt hierarchical edge computing system, which runs object detection on 1st edge node and running motion tracking on 2nd edge node), which is embedded with an ARM A56 CPU and NVIDIA Maxwell GPU. The 1st edge is equipped with one IP camera that is connected through USB. The 2nd edge node has a more powerful GPU, such as the GeForce RTX 2080 SUPER GPU. We implemented an Android application for sending an alarm. For the overall implementation, we implemented all the modules using Python and TensorFlow.

**1st edge node:** It includes object detection and communication with the 2nd edge node. For object detection, we implemented the frame difference (Equation (Equation 3)) using the OpenCV library (version 4.2.0) [30]. Herein, the YOLO network is used for object detection [27] with a frame that exceeds the threshold of the frame difference. Particularly, here, the 1st edge node adopted YOLOv3-tiny, which is a very small model for constrained environments. It uses Python sockets to deliver frames to the 2nd edge node.

**2nd edge node:** It includes motion tracking, control module, and communication with both the 1st edge node and cloud server. Basically, it implements the control module using Python Thread class to manage the data from each 1st edge node. The motion tracking module, which stops the process to reduce GPU memory resources, is implemented using Python Process class. In the control thread, the communication and control motion tracking process is implemented as mentioned above (Section 2.3). First, it receives frames with messages from the 1st edge node using Python socket, and then it puts the frames into a queue that allows communication with the motion tracking process. Then, it measures the time that the queue is empty (i.e., the time when the 1st edge node does not detect the object). If the measured time is greater than the predefined threshold, it pushes the trigger message to the queue to stop the motion tracking procedure. For motion tracking, we use tf-pose-estimation [31], which is a human pose estimation algorithm implemented using TensorFlow. Then, the 2nd edge node sends a message to the cloud server using an HTTP message.

**Cloud server:** We implemented an Android application and a web page for the users. The cloud server hosts a web page that shows the view of the camera at the 1st edge node. The web page was implemented using Flask with Python [32]. Furthermore, we implemented an Android application using WebView [33], which displays web pages on user devices. In addition to providing camera images, the cloud sends a push alarm to authorized users on detecting abnormal behaviors of people. To send push alarms to authorized users, it uses the Firebase Cloud Messaging [34] solution with a database that stores user IDs and the token of the Android device according to the 1st edge node ID.

Figure 5 presents the implementation results for each node. Figure 5a presents the screen on the 1st edge node, 2nd edge node and cloud server from the left to the right. Users can see the video on the web page and receive a push alarm with Android phone. In addition, Figure 5b shows the screen for performing the YOLOv3-tiny algorithm on the 1st edge node, and Figure 5c shows the screen for performing the tf-pose-estimation on the 2nd edge node.

## 4. Performance Evaluation

In this section, we provide measurement-based performance evaluations to verify the effectiveness of the proposed AdaMM framework. The benchmark and the proposed framework for evaluation are briefly described as follows:**Benchmark**: For the comparison, the always-on motion tracking process without the proposed AdaMM is benchmarked. We denote this benchmark as Baseline.-**Baseline**: The Baseline runs hierarchical object detection and motion tracking, which follows a framework architecture that is the same as the one in the proposed AdaMM framework for fair evaluation. Here, if the motion tracking model is loaded for motion tracking, then the Baseline continues motion tracking regardless of the presence of an object, while retaining the GPU memory for the loaded model.**Proposed**: The proposed AdaMM framework can have three variants: (1) Proposed AdaMM, (2) proposed AdaMM without frame differencing, and (3) proposed AdaMM without adaptive process management.-**Proposed AdaMM**: This includes all proposed modules, such as frame differencing and adaptive process management.-**Proposed AdaMM without frame differencing**: This does not have a frame differencing function at the 1st edge node. Consequently, the 1st edge node continues running the object detection module regardless of any frame difference.

For extensive measurement-based evaluation, we utilize the commercial edge device, Jeston Nano, as the 1st edge node. It has a 128-core Maxwell GPU. For the 2nd edge node, we use a desktop with a GeForce RTX 2080 SUPER. Additionally, we prepare two sample videos, video1 and video2, to simulate the intermittent object appearance scenario in video surveillance. The values of the length, frame rate, and resolution of each video are approximately 300 s, 30 fps, and 1520 × 720, respectively. The distribution of object appearance in both the videos is presented in Figure 6. Here, the values of 1 and 0 indicate that the object is detected or undetected, respectively, by the 1st edge node. In video1, the object is detected when between 0 s and 60 s, but not when between 61 s and 170 s. On the contrary, in video2, the object always exists during the entire playback time.

The framework configuration and parameters are summarized in Table 2. We consider two important performance metrics measured on the 2nd server: (1) GPU utilization, denoted as $GPU_{u}$, and (2) GPU memory utilization, denoted as $GPU_{m}$, where the range of both $GPU_{u}$ and $GPU_{m}$ is [0%, 100%]. To load the model for motion tracking, approximately 3600 MiB (total memory 7979 Mib) is required; therefore, *m* is approximately 45%. GPU utilization *u* is approximately 30% while loading the model and approximately 10% while tracking object movement and motion because it does not require to load the model. *X* is the probability of object appearance, which is calculated by the total elapsed time of each object appearance in the video/total playback time, and *T* is the average elapsed time of each object appearance. For instance, when the length of the video is 300 s with X=10% and T=10 s, the average elapsed time of each object appearance can be estimated to be 10 s from T=10 s, and the total elapsed time of each object appearance can be estimated to be 300 s ×X(=0.1) = 30 s. Further, we set $\theta_{m}$ = 10 s, 30 s on the 2nd edge node.

### 4.1. Evaluation of the Proposed Framework: Specific Scenario

Figure 7 presents the performance of the proposed AdaMM framework without frame differencing compared to Baseline. Here, we consider two 1st edge nodes that deploy the proposed AdaMM without frame differencing and the Baseline. In addition, both the 1st edge nodes receive the video1 in Figure 6, and, after 60 s, they receive video2. Subsequently, they request a motion tracking service to the 2nd edge node. Specifically, at 0 s, both frameworks load model 1 for the video1 request, and at 60 s, load model 2 for the video2 request. However, the proposed framework without frame differencing uses less GPU memory than the Baseline for all $\theta_{m}$ (threshold for stopping process). This is because the proposed framework releases the GPU memory for the model, but the Baseline holds it regardless of whether the object is detected or not for $\theta_{m}$ s. In the proposed framework, GPU memory utilization of the 2nd edge node varies according to $\theta_{m}$. In the case of $\theta_{m}=10$ (Figure 7a), the proposed framework stops the motion tracking procedure twice by releasing the GPU memory. This is because it does not receive any frame from the 1st edge node during 10 s ($\theta_{m}$ value) when at approximately 60 s and 240 s. Here, the proposed framework immediately reloads the model for motion tracking at approximately 251 s because it receives frames from the 1st edge node. This incurs additional latency for the reloading model, causing an overhead. While reloading the model, the proposed framework cannot run motion tracking even though there is an object appearance, which is a trade-off between saving GPU memory and latency. Accordingly, $\theta_{m}$ is carefully determined depending on the application, which is beyond the scope of this paper. In the case of $\theta_{m}=30$ (Figure 7b), the proposed framework stops the procedure once because, at approximately 240 s, it receives a frame within $\theta_{m}$.

### 4.2. Impact of Object Detection Probability on Performance

To understand how the probability of object appearance affects the performance, we represent the GPU usage utilization and memory utilization with respect to *X* and *T* (in the case of $\theta_{m}=10$). To this end, we manipulate the video to have varying *X* and *T* values. Figure 8, Figure 9 and Figure 10 present the results, where the 2nd edge node receives and processes the video with X% probability of object appearance. The Baseline framework does not release GPU memory after loading the model even if the GPU usage is nearly 0% for all *T* and *X*, which means that the 2nd edge node does not run motion tracking. Consequently, the performance of the Baseline is independent of *T* and *X*. On the contrary, in the proposed framework, the performance is affected by *T* and *X* because the GPU memory is released if the 2nd edge node does not receive the frame from the 1st edge node within $\theta_{m}$ (i.e., 10 s in this test). Specifically, Figure 8 shows the result based on *T* when *X* is 10%. Here, the smaller the value of *T*, the more the likelihood of frequent object appearance within $\theta_{m}$. Similar to the previous evaluation in Figure 7, the proposed framework can reduce GPU memory utilization compared to Baseline because the Baseline holds the memory for motion tracking, but the proposed framework releases it when the object is rarely detected. Similar to Figure 8, in Figure 9, the proposed framework can also reduce GPU memory utilization compared to Baseline. However, if the 2nd edge node receives a frequent frame during GPU memory release (in the case of T=10), a service delay is caused for reloading the model (e.g., 3 s). This reduces the accuracy of motion tracking owing to the frame drop when the 2nd edge node receives a frame as soon as the motion tracking procedure is stopped. However, if *T* is sufficiently large, this risk can be mitigated. It can be seen in Figure 11b, that, when *T* is 30, the 2nd edge node saves GPU memory each about 32 s. However, in Figure 10, the object is detected frequently (X=70%); therefore, the proposed framework incurs service delay while it does not have any chance to reduce GPU memory utilization.

As shown in Figure 11a, we evaluate the proposed framework based on a Monte Carlo simulation with 100 random cases (random distribution of object appearance). Here, the light shading in the graph depicts additional GPU memory utilization for holding GPU memory regardless of object appearance in the Baseline. We can easily conclude that the proposed framework requires less GPU memory than the Baseline over varying *X* and *T*.

The proposed framework reduces the GPU memory for all *X* as *T* increases. The reason why the frequency of holding and waiting time ($\theta_{m}$) decreases for releasing GPU memory as the average time object stays increase.

In the case of X=10% when T=10, the object can appear three times, 10 s each. In this case, the 2nd edge node holds the GPU memory for a total of approximately 60 s, which is the summation of the times for object movement, motion tracking, and waiting time ($\theta_{m}$). However, when T=30, the object appears one time, 30 s; in this case, the 2nd edge node holds the GPU memory for a total of approximately 40 s. In the case when X=10%, the proposed framework reduces the GPU memory by approximately 76.8%, 78.5%, and 78.0% over various *T* compared to Baseline. Even in the case of X=30%, the proposed framework provides better performance than the Baseline by reducing the GPU memory by approximately 46.21%, 50.83%, and 54.47% over various *T*. Even if there are frequent object appearances, such as X=70%, the proposed framework reduces the GPU memory by approximately 11.64%, 14.49%, and 16.25% over various *T* compared to Baseline. From these observations, we can conclude that the performance gain of the proposed framework increases as the value of *X* decreases and *T* increases. In other words, the proposed framework achieves better performance when the object is rarely detected. Efficiency of the proposed framework is indicated by the average GPU release time, which is the total GPU release time per number of GPU memory release on the 2nd edge node, as shown in Figure 11b. Specifically, the average GPU release time over *T* in the case of X=10%, is 71, 97, and 129 s. In the case of X=30%, it is 27, 35, and 62 s. Lastly, in the case of X=70%, it is 13, 18, and 24 s. In other words, as *X* decreases, the object is rarely detected, and the average GPU release time increases. Moreover, as *T* increases, the average GPU release time increases, but the variance also increases.

### 4.3. Frame Differencing

Finally, the impact of frame differencing on the average GPU memory utilization for the proposed AdaMM is presented in Figure 12. We evaluated two cases: (1) the proposed AdaMM and (2) the proposed AdaMM without frame differencing. Here, it can be seen that frame differencing has an impact on the reduction of GPU usage utilization. Specifically, with frame differencing, the proposed AdaMM further reduces the GPU usage utilization by 21%. This is because the 1st edge node rarely performs the object detection module by filtering similar frames using frame difference; therefore, the 2nd edge node has a similar effect to that when object detection rarely occurs, as mentioned above.

## 5. Conclusions

In this study, we designed the AdaMM framework that adaptively performs object movement and motion tracking in a hierarchical edge computing system to save GPU memory. Specifically, such GPU memory saving is beneficial for accommodating more AI services on the edge computing server with limited GPU memory. We implemented and experimented the framework on commercial edge devices (e.g., Jetson Nano). Measurement-based performance evaluation reveals that the proposed AdaMM can save GPU memory utilization up to 78.5% compared to the benchmark. Furthermore, performance of the proposed AdaMM is affected by the probability of object appearance (*X*) and threshold for the stopping process ($\theta_{m}$). The proposed AdaMM framework can achieve the best performance improvement when an object is rarely detected. Additionally, regarding the threshold variable ($\theta_{m}$), when the threshold value is significantly large, the GPU memory cannot be released at the right time; therefore, system performance cannot be improved. However, if the threshold value is too small, the GPU memory is released frequently, resulting in additional overhead. As future work, we can overcome the limitation of our proposed framework that uses a fixed threshold by dynamically adjusting the threshold variable by predicting the probability of object appearance video. Moreover, in the AdaMM framework, 1st edge node can reduce energy consumption since it does not perform unnecessary object detect by using frame differencing. Accordingly, we will also analyze energy efficiency of overall proposed system as another future work.

## Figures and Tables

**Figure 1 sensors-21-04089-f001:**
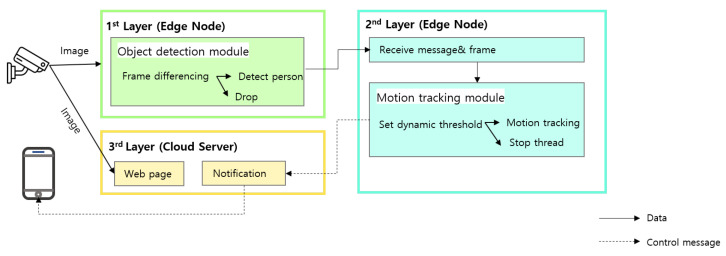
Model of the proposed framework.

**Figure 2 sensors-21-04089-f002:**
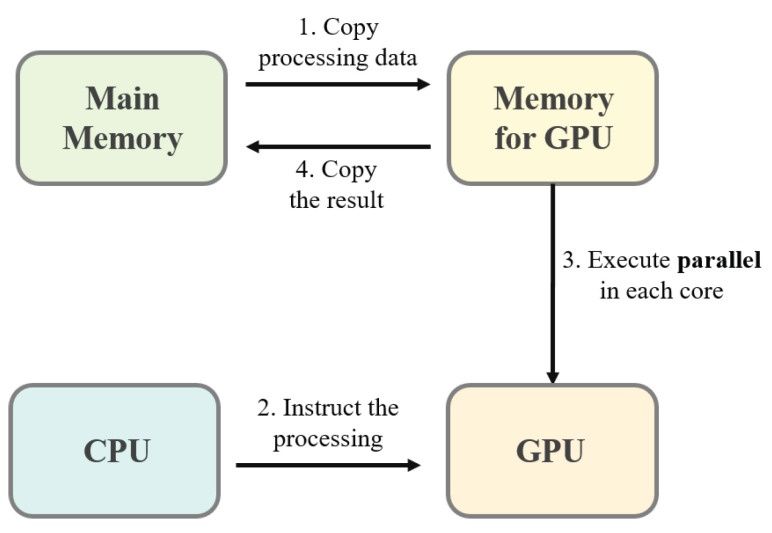
Processing flow on CUDA.

**Figure 3 sensors-21-04089-f003:**
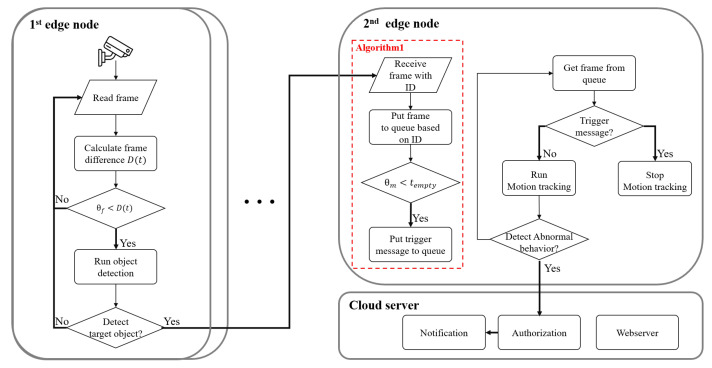
Flow diagram of the proposed AdaMM framework with multiple 1st edge nodes.

**Figure 4 sensors-21-04089-f004:**
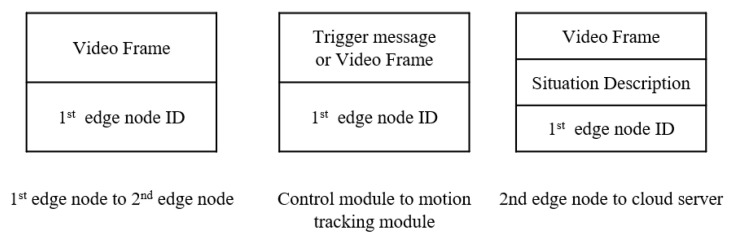
Message format.

**Figure 5 sensors-21-04089-f005:**
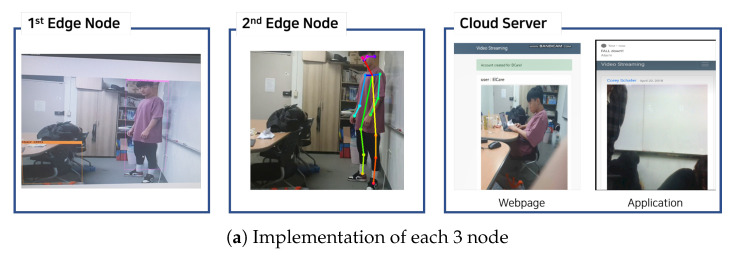
Result of implementation on 1stedge node, 2nd edge node, and cloud server.

**Figure 6 sensors-21-04089-f006:**
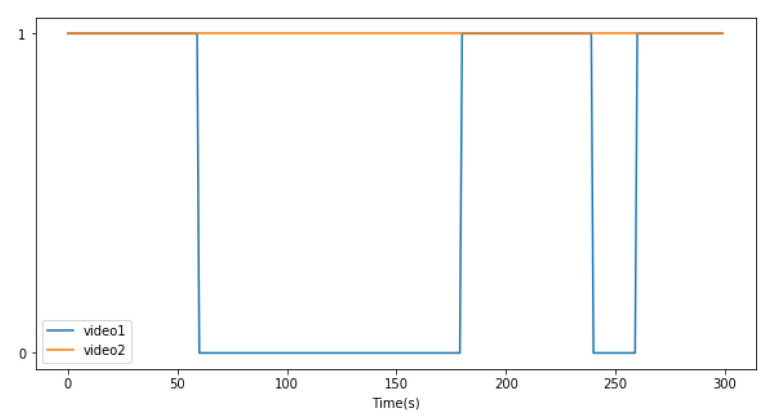
Video information.

**Figure 7 sensors-21-04089-f007:**
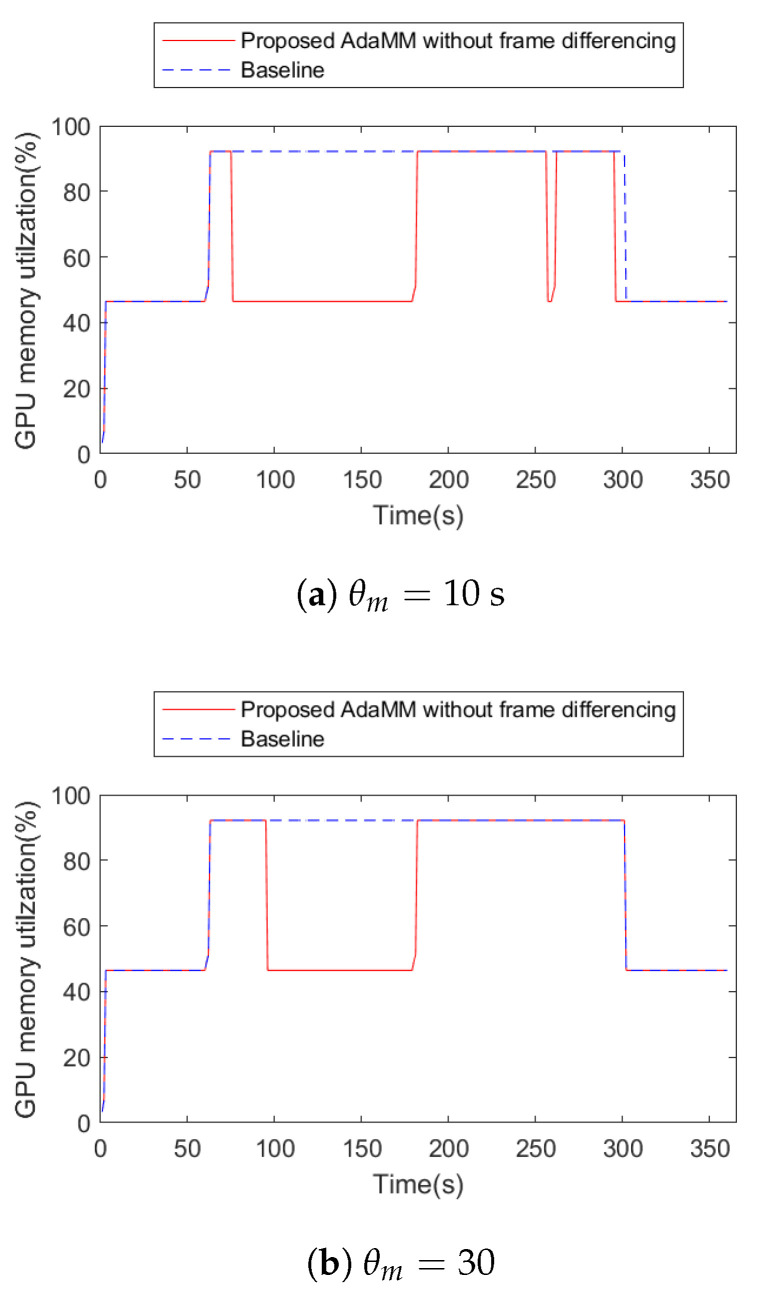
GPU memory utilization on 2nd edge server with proposed AdaMM framework compared to Baseline framework.

**Figure 8 sensors-21-04089-f008:**
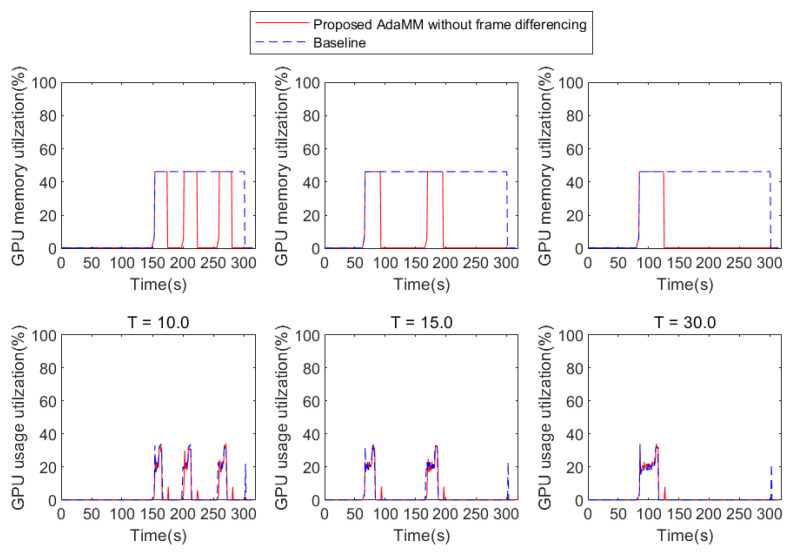
GPU usage and memory utilization based on *T* when *X* = 10% (in the case of $\theta_{m}=10$).

**Figure 9 sensors-21-04089-f009:**
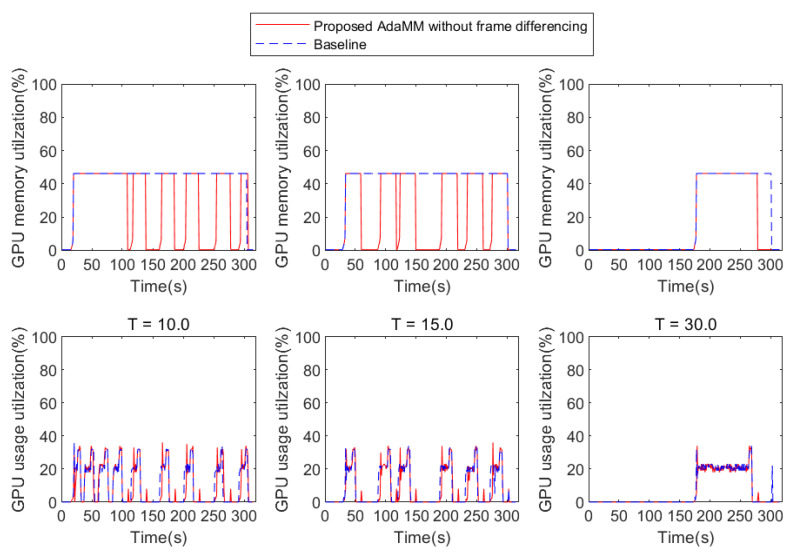
GPU usage and memory utilization based on *T* when *X* = 30% (in the case of $\theta_{m}=10$).

**Figure 10 sensors-21-04089-f010:**
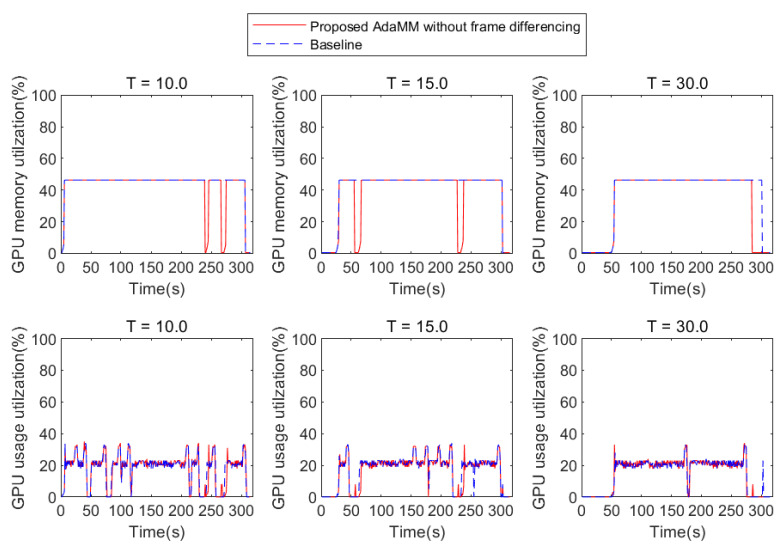
GPU usage and memory utilization based on *T* when *X* = 70% (in the case of $\theta_{m}=10$).

**Figure 11 sensors-21-04089-f011:**
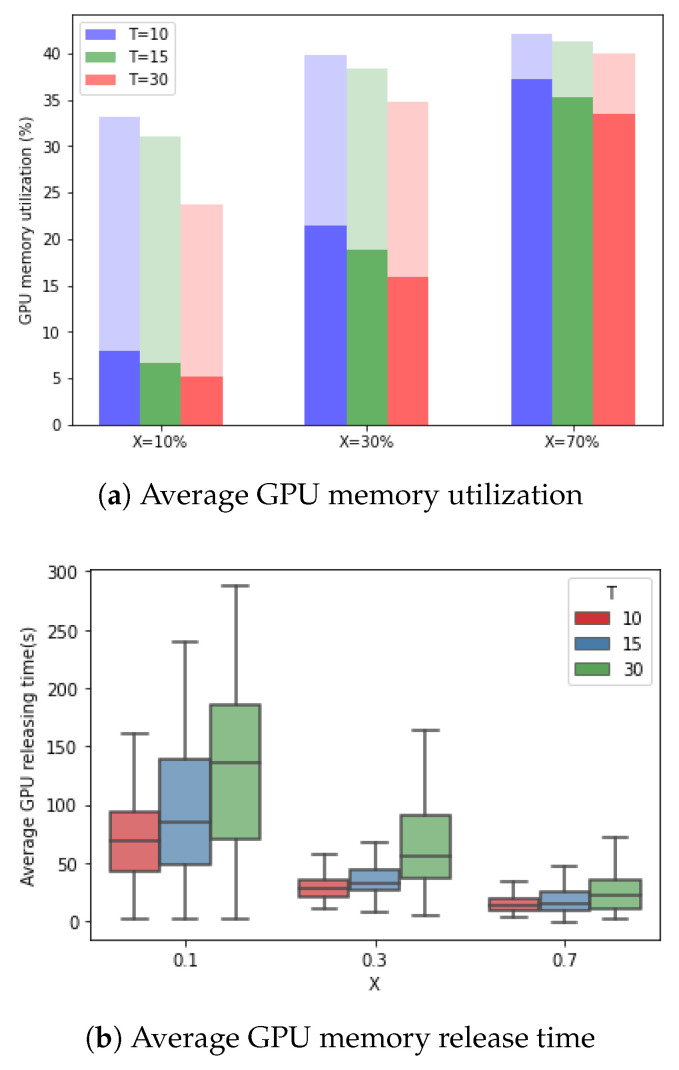
Average GPU memory utilization and average GPU memory release time with respect to *T* and *X* on the Baseline and proposed AdaMM without frame differencing. Average GPU memory utilization of the proposed framework (dark shading) and additional GPU memory utilization of Baseline (light shading).

**Figure 12 sensors-21-04089-f012:**
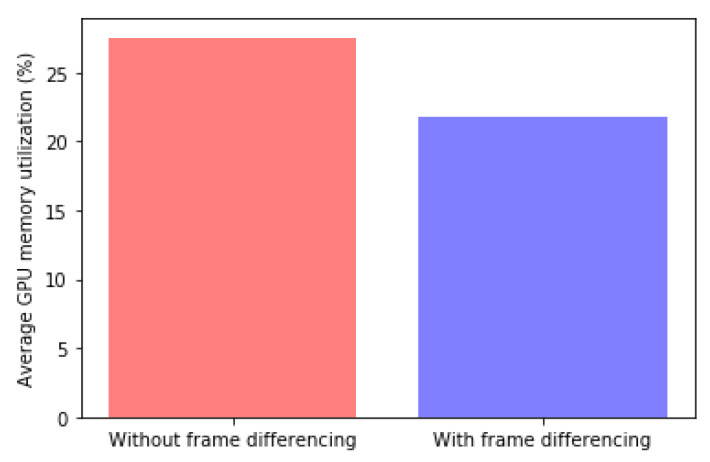
Impact of frame differencing on average GPU memory utilization. Average GPU memory utilization of AdaMM (dark shading) and additional GPU memory utilization of Baseline (light shading).

**Table 1 sensors-21-04089-t001:** Summary of major symbols.

Symbol	Definitions
f(x,y,R)	Red channel pixel values in specific coordinates (x,y)
f(x,y,G)	Green channel pixel values in specific coordinates (x,y)
f(x,y,B)	Blue channel pixel values in specific coordinates (x,y)
$gray_{t}\left(x,y\right)$	Gray scale value in specific coordinates (x,y) at time *t*
ϕ	Threshold for frame differencing
$\theta_{f}$	Threshold for triggering object detection
$\theta_{m}$	Threshold for stopping the process
d(t)	Whether the absolute difference of pixel between *t* and t−1 is exceed the threshold ϕ
D(t)	Frame difference at time *t*

**Table 2 sensors-21-04089-t002:** System configuration and parameters.

Parameter	Value
GPU utils, $GPU_{u}$ (%)	[0%, 100%]
GPU memory usage, $GPU_{m}$ (%)	[0%, 100%]
Probability of object appearance, *X* (%)	10%, 30%, 70%
Average time the object stays in the video, *T* (s)	10 s, 15 s, 30 s
Threshold for stopping the process, $\theta_{m}$ (s)	10 s, 30 s

## Data Availability

Data sharing not applicable.

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
