# Peer review of "AdaMM: Adaptive Object Movement and Motion Tracking in Hierarchical Edge Computing System"

_sensors, 2021, doi:10.3390/s21124089_

Round 1
Reviewer 1 Report
This paper presents a novel adaptive object movement and motion tracking (AdaMM) framework in a hierarchical edge computing system for achieving GPU memory footprint reduction of deep learning (DL)-based video surveillance services. The proposed framework aims to adaptively release the unnecessary standby object motion and movement tracking model to save GPU memory by utilizing light tasks such as frame difference calculation and object detection in a hierarchical manner. Designed experiments show the efficiency and effectiveness of the proposed design. This paper is relatively well organized, however it is not qualified for the publication before a minor revision. Comments are listed below.
1) This work sounds like the concept of federal learning. Well address the novelty different from other famous techniques.
2) From Figure 1, it seems the proposed framework has three not obvious layers. For each layer, please explain the differences and capabilities clearly.
3) Shrink some figures (e.g., Figure 2) to present more professionally.
4) It is better to Replace the clipped screenshot figures with clear vector ones.
5) For the evaluation, authors used the YOLOv3-tiny for testing, if necessary, explain the reason why the latest YOLO is not used.
6) A comprehensive symbol table is needed to introduce all used math notations for easy understanding.
7) Make the References more comprehensive, the current References is inadequate and messy (double check whether the right template formatting is applied). Besides the object movement and motion tracking, the similar work may can be applied in some other promising scenarios (e.g., Blockchain, Big Data or other IoT systems). If the above related work can be discussed, it can strongly improve the research significance. For the improvement, the following papers can be considered to make the references more comprehensive.
- Park and Y. Kim, “User profile system based on sentiment analysis for mobile edge computing,” Computers, Materials & Continua, vol. 62, no. 2, pp. 569–590, 2020.
- Yan, Y. Dai, Z. Zhou, W. Jiang and S. Guo, “Edge computing-based tasks offloading and block caching for mobile blockchain,” Computers, Materials & Continua, vol. 62, no. 2, pp. 905–915, 2020.
- Dai, J. Yi, Y. Zhang and L. He, “Multi-scale boxes loss for object detection in smart energy,” Intelligent Automation & Soft Computing, vol. 26, no.5, pp. 887–903, 2020.
- Lee, H. Ahn, H. Ahn and S. Lee, “Visual object detection and tracking using analytical learning approach of validity level,” Intelligent Automation & Soft Computing, vol. 25, no.1, pp. 205–215, 2019.
- Gumaei, M. Al-Rakhami and H. AlSalman, “Dl-har: deep learning-based human activity recognition framework for edge computing,” Computers, Materials & Continua, vol. 65, no. 2, pp. 1033–1057, 2020.
- Peng and Q. Li, “Research on the automatic extraction method of web data objects based on deep learning,” Intelligent Automation & Soft Computing, vol. 26, no.3, pp. 609–616, 2020.
Cen Chen, Kenli Li, Aijia Ouyang, Zhuo Tang, Keqin Li: GPU-Accelerated Parallel Hierarchical Extreme Learning Machine on Flink for Big Data. IEEE Trans. Syst. Man Cybern. Syst. 47(10): 2740-2753 (2017)
Wangdong Yang, Kenli Li, Zeyao Mo, Keqin Li: Performance Optimization Using Partitioned SpMV on GPUs and Multicore CPUs. IEEE Trans. Computers 64(9): 2623-2636 (2015)
Reviewer 2 Report
The paper is an interesting presentation of CCTV based edge computing system with multi-layered nodes, which can redduce the computing load of the units, especially motion tracking camera. Two minor improvement would be fine:(1) If the multi-layered structure and trigering mechanism are unique as you know. As a distributing sysem this mechansim option seems very reasonable. The uniqueness and novelty of the approach could be furture clearified;(2) What is the meaningness of reducing main memory cost for the 2nd edge node? You can say the tracking camera will begin to track motion of objects if only something is moving so that energy could be saved, for tracking takes electricity.
Reviewer 3 Report
The paper presents an adaptive object movement and motion tracking (AdaMM) for achieving GPU memory footprint reduction of DL-based video surveillance services. The authors describe analytically the framework, the implementation as well as the results. Specifically, the results are depicted also in numerous Figures which are commented in details. The manuscript ends with a conclusions section and the references list is rich.
Reviewer 4 Report
In this paper, the authors proposed a hierarchical edge computing system for object tracking and movement.
The paper is very well written and well presented. In my opinion, only these two things should be fixed:
- To have a better overview of the system performance, it could be useful to have the results obtained by using only the jetson nano;
- In line 388, there is a wrong reference to a figure ( Fig ??).
